# ESCA: Enabling Seamless Codec Avatar Execution through Algorithm and Hardware Co-Optimization for Virtual Reality

**Mingzhi Zhu**[1,3]    **Ding Shang**[1]    **Sai Qian Zhang**[1,2]

[1]Tandon School of Engineering, New York University
[2]Courant Institute of Mathematical Sciences, New York University
[3]Rensselaer Polytechnic Institute
{mingzhi.zhu,dingshang,sai.zhang}@nyu.edu

## Abstract

Photorealistic Codec Avatars (PCA), which generate high-fidelity human face renderings, are increasingly being used in Virtual Reality (VR) environments to enable immersive communication and interaction through deep learning–based generative models. However, these models impose significant computational demands, making real-time inference challenging on resource-constrained VR devices such as head-mounted displays (HMDs), where latency and power efficiency are critical. To address this challenge, we propose an efficient post-training quantization (PTQ) method tailored for Codec Avatar models, enabling low-precision execution without compromising output quality. In addition, we design a custom hardware accelerator that can be integrated into the system-on-chip (SoC) of VR devices to further enhance processing efficiency. Building on these components, we introduce *ESCA*, a full-stack optimization framework that accelerates PCA inference on edge VR platforms. Experimental results demonstrate that ESCA boosts FovVideoVDP quality scores by up to $+0.39$ over the best 4-bit baseline, delivers up to $3.36\times$ latency reduction, and sustains a rendering rate of 100 frames per second in end-to-end tests, satisfying real-time VR requirements. These results demonstrate the feasibility of deploying high-fidelity codec avatars on resource-constrained devices, opening the door to more immersive and portable VR experiences. Paper website can be found at `https://zmzfpc.github.io/ESCA/`.

## 1 Introduction

Photorealistic telepresence [29, 41] in VR requires real-time transmission and rendering of facial expressions with lifelike fidelity. Photorealistic Codec Avatars (PCA) have emerged as a promising solution by leveraging variational autoencoder (VAE) models to compress and reconstruct human faces for remote interactions [30]. In this framework, an inward-facing camera on the sender's VR device captures the user's facial expressions, and an encoder generates a compact latent representation. This latent code is transmitted wirelessly to the receiver, where a decoder reconstructs a high-quality facial image and 3D avatar. Although the pipeline supports efficient streaming of photorealistic facial data, achieving the combination of high visual fidelity and ultra-low latency on mobile VR hardware continues to pose a significant technical challenge [33].

A primary source of processing latency in Codec Avatars is the decoder network, which relies heavily on transposed convolution layers to synthesize high-resolution facial images. While effective for generating high-quality visuals, these layers are computationally demanding and introduce significant

39th Conference on Neural Information Processing Systems (NeurIPS 2025).

latency. This poses a major obstacle to real-time performance, as delivering a seamless user experience typically requires sustaining 90 frames per second [14, 18].

To address this bottleneck, neural network quantization has been widely explored in prior work as a means to enable low-latency execution of deep models [9, 43, 5, 27, 8]. However, two key challenges arise in the context of Codec Avatars: First, due to the large scale of these networks, applying quantization-aware training (QAT) across the entire model is to train computationally impractical. Second, activation outliers greatly exacerbate quantization errors, especially in low-precision regimes, resulting in pronounced degradation of visual quality. This issue is most evident in transposed convolution layers, where stochastic latent codes and the absence of normalization induce long-tailed activation distributions with pronounced spikes, as illustrated in Figure 1. Because these outliers dominate the value range, they diminish quantization precision and cause frame-dependent errors. This manifests as temporal artifacts in the reconstructed facial image, including flickering, checkerboard patterns, and unstable shading [38].

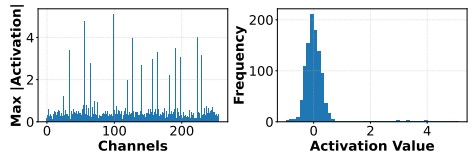

Recent advances have proposed effective strategies for managing outliers in low-bit inference of large models, such as activation smoothing [52] and weight rotation [3, 51]. These approaches enable 4-bit precision in large language models by rescaling channels or rotating weight matrices to suppress extreme values while preserving accuracy. However, such strategies cannot be directly applied to the PCA model. The presence of transposed convolution layers and non-linearities prevents straightforward use of offline weight rotations or channel scaling, as these modifications would alter the model's outputs. Consequently, existing outlier-smoothing techniques are incompatible with the PCA decoder architecture, leaving efficient 4-bit quantization for high-fidelity facial generation as an open challenge.

Figure 1: The left panel shows the maximum activation value of each channel of a sample input. The right part shows the aggregated activation distribution over all spatial locations and channels.

Beyond algorithmic challenges, VR HMDs remain resource-constrained, which makes execution of the computationally intensive PCA decoder slow. While some devices incorporate GPUs and NPUs [34], these units must also support concurrent tasks such as rendering [32], image processing [36], and other AI workloads [48, 26, 16]. Addressing this constraint calls for a dedicated hardware accelerator integrated as a plug-in module within the SoC to manage Codec Avatar inference. Yet, the decoder's reliance on transposed convolution layers poses a unique difficulty, as their intrinsic structured sparsity severely limits accelerator utilization and efficiency.

To overcome these challenges, we propose a comprehensive quantization and acceleration framework for Codec Avatars, enabling real-time inference on resource-constrained VR devices. Our approach introduces several novel techniques to maintain visual quality under low-bit quantization and a co-designed hardware solution to meet strict latency requirements. In summary, our contributions are:

- Input Channel-wise Activation Smoothing (ICAS): We introduce a novel input channel-wise smoothing module inserted during training to alleviate extreme inter-channel activation disparities in the VAE decoder. By reducing outlier activations, ICAS diminishes quantization error and prevents aberrations when the model is later quantized to low bit-widths.

- Facial-Feature-Aware Smoothing (FFAS): We develop a region-aware smoothing strategy that uses facial masks to identify key areas like the eyes and mouth. Based on the activation variance in these regions, FFAS selectively skips smoothing for the channels most critical to fine details, preserving important textures while still smoothing less sensitive regions.

- UV-weighted Hessian-Based Weight Quantization: We propose a weight quantization scheme guided by a UV-mapping weighted Hessian. This method computes second-order sensitivity and weights it by the UV importance of each face region, thereby concentrating on the precision of weights that most affect critical facial features.

- Customized Hardware Accelerator: We co-design a specialized hardware accelerator to support our quantized Codec Avatar model with high-throughput 4-bit and 8-bit operations. The accelerator features an input-combining mechanism to exploit the structured sparsity of the activation matrix. Moreover, an optimized end-to-end pipeline is applied to deliver over 100 FPS inference on an VR headset ensuring smooth, real-time avatar rendering.

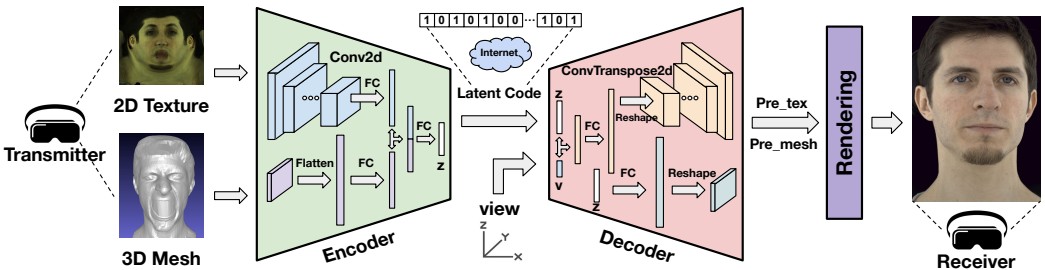

Figure 2: VAE Framework of Codec Avatar models.

## 2 Background and Related Works

### 2.1 Codec Avatar and Photorealistic Pipeline

PCA are neural face models that enable authentic telepresence in VR by rendering lifelike 3D avatars of users in real-time [21, 30]. They are typically implemented as a variational autoencoder (VAE) [29] framework that transmits facial expressions efficiently between users. Figure 2 illustrates the overall stucture of Codec Avatar models. This VAE-based approach has been demonstrated on VR headsets (e.g. Meta Quest Pro) as a feasible solution for real-time face-to-face communication, achieving a convincing sense of social presence while drastically reducing transmitted data compared to raw video [33].

Figure 3 (a) decomposes the end-to-end execution pipeline of the Codec Avatar application, which can be divided into five main stages: Sensing, Encoding, Transmission, Decoding, and Rendering. As illustrated in Figure 3 (b), the VR device comprises several hardware modules, primarily the CPU, GPU, front-facing camera, and memory subsystem. To gauge the limits of current commercial platforms, we profile the Snapdragon XR2 Gen 2 SoC [39] powering Meta's Quest 3. Running the full Codec Avatar model on Qualcomm AI Hub [40] yields a median inference latency of 39.6 ms, only 25.25 FPS, even before accounting for Sensing, Transmission, and Rendering overheads. Offloading to cloud servers is likewise untenable due to added latency and privacy concerns [33]. Thus, to enable truly immersive telepresence, Codec Avatar must execute on-device within the available compute budget [46, 7]. Moreover, running the PCA module continuously consumes additional VR hardware resources, leaving fewer computational resources for other applications to operate effectively. This motivates the development of custom hardware accelerator as a plug-in of the VR SoC to handle Codec Avatar decoding. Prior works have shown that specialized architectures for generative models can vastly improve efficiency [20]. For instance, an FPGA-based transposed-convolution engine achieved up to $3\times$ better performance-per-watt than a GPU on similar tasks [4]. In summary, a full-stack approach that co-designs the Codec Avatar with hardware support is essential to reach 90 FPS real-time photorealistic telepresence [14, 18] on power-constrained VR devices.

### 2.2 Post-Training Quantization

Post-training quantization (PTQ) [35, 12, 42, 52, 3, 22, 28, 51] converts a pre-trained floating-point network to low-precision integers without retraining, making it an attractive deployment strategy for resource-constrained VR hardware. Classic weight-only PTQ schemes such as AdaRound [35], GPTQ [12], and OmniQuant [42] minimize layer-wise reconstruction error by solving local optimization problems, while recent activation-aware methods further suppress outliers to unlock 4-bit inference for language models. SmoothQuant [52] migrates per-channel activation magnitude into the weights via learned scaling factors, where QuaRot [3], DuQuant [22] and SpinQuant[28] apply orthogonal rotations to jointly smooth activations and weights in a Hessian-aware manner. Given $Y = XW$, they insert a matrix $R$ where $R^\top R = I$ and rewrite the product as $Y = XW = (XR)(R^\top W)$, thereby smoothing $XR$ without changing the network output. The new weight $R^\top W$ is folded offline, while the rotated activation is absorbed into the preceding layer, so inference cost is unchanged.

These techniques are effective because Transformer layers [47] are dominated by matrix multiplications, whose linear properties remain intact under such transformations. However, directly applying these methods to convolution-based codec-avatar decoders is non-trivial. First, convolutional generators utilize 4-D kernels and 3-D activations, violating the 2-D matrix assumptions that

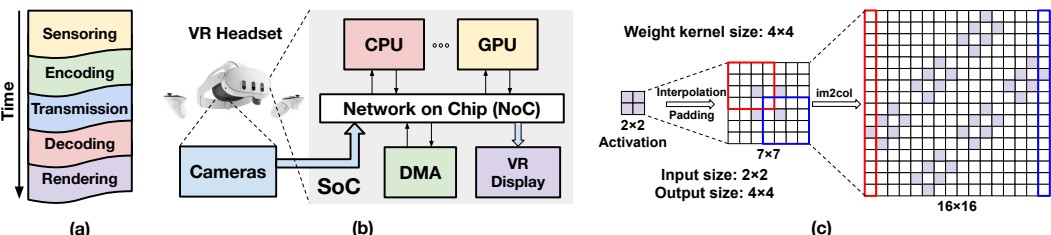

Figure 3: (a) Execution pipeline of the entire Codec Avatar system. (b) Architecture of normalized VR headset SoC. (c) Illustration of transposed convolution. Purple squares represent non-zero activation, and white squares represent zero activation.

underpin SmoothQuant's channel-wise scaling. Second, our decoder extensively employs transposed convolutions followed by non-linear activations (e.g., LeakyReLU [53]), rendering any offline weight rotation invalid once activations pass through these non-linear transformations. In summary, existing PTQ methods are ill-suited to the architectural and signal-processing peculiarities of codec-avatar decoders, leaving efficient 4-bit quantization of high-fidelity face generators an open problem.

## 2.3 Transposed Convolution and Im2col Transformation

Modern Codec Avatar decoders rely on transposed convolution (ConvTranspose) layers, also known as deconvolutions, to upsample low-resolution feature maps into high-resolution images or textures [29, 30, 13]. The transposed convolution expands the spatial dimensions by inserting zeros between and around input pixels before applying a convolution kernel, effectively spreading the feature map. The output width W' of the activation after this pre-processing can be computed as:

$$W' = W + 2(K - P - 1) + (W - 1)(S - 1) \tag{1}$$

where $W$ denotes the original width of the activation map, and $K$, $P$, and $S$ represent the kernel size, padding, and stride, respectively. The activation maps and kernels are assumed to be square, meaning the width and height are equal. For example, considering the first layer of the decoder, the activation width is W = 2 and K = 4, S = 2, P = 1. According to the equation, the activation width after inserting zero becomes $2 + 2 \times (4 - 1 - 1) + (2 - 1) \times (2 - 1) = 7$. As illustrated in Figure 3 (c), one zero is inserted between every adjacent activation (interpolation), and two layers of zeros are padded around the boundary. Thus a $2 \times 2$ feature map turns into $7 \times 7$ after applying zero-inserting.

To leverage the high throughput of a systolic-array-based accelerator, the convolution operations are typically transformed into general matrix-matrix multiplication (GEMM) [11]. This requires flattening high-dimensional activations and weights into two-dimensional matrices using the im2col transformation [6, 1]. However, due to the zero-inserting introduced during transposed convolution, the resulting im2col-transformed activation matrix becomes extremely sparse. As shown in Figure 3 (c), this manifests as a checkerboard-like sparsity pattern, with more than 85% of the elements being zero. This high degree of sparsity significantly degrades the efficiency of the hardware accelerator [20], as the majority of multiply-accumulate (MAC) operations become redundant (multiplication by zero), wasting both compute cycles and memory bandwidth [54].

## 3 Methods

Our pipeline tackles the twin challenges of low-bit quantization and real-time deployment of Codec Avatar decoders through four tightly-coupled components: (a) Channel-wise Activation Smoothing; (b) Facial-Feature-Aware Smoothing; (c) UV-Weighted Post-Training Quantization; (d) Input-combined DNN hardware accelerator. Together, the techniques shown in Figure 4 and 5 deliver artifact-free 4/8-bit inference while boosting avatar-decoder throughput on our prototype accelerator.

## 3.1 Input Channel-wise Activation Smoothing

We introduce Input Channel-wise Activation Smoothing (ICAS) to equalize the scale of activation across channels in each transposed convolution which aims to reduce the quantization difficulty of activations. Our method is inspired by SmoothQuant [52], which migrates the quantization difficulty

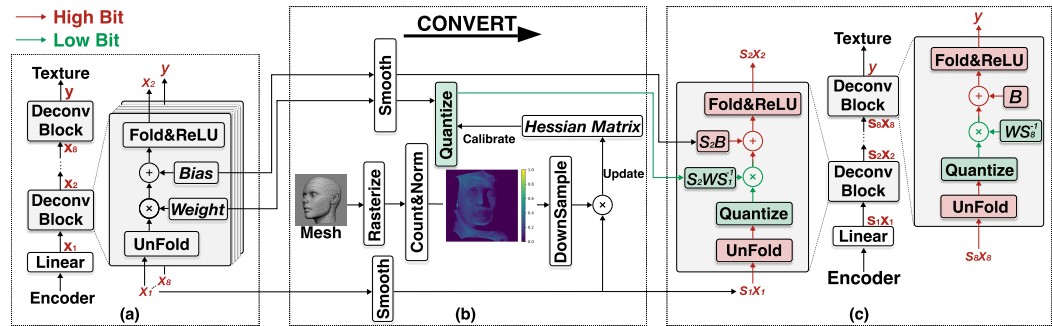

Figure 4: Convert the Codec Avatar decoder to a quantized model. The pipeline consists of three main components: (a) The original Codec Avatar decoder, (b) the UV-Weighted Post-Training Quantization, and (c) the decoder layer after Input Channel-wise Activation Smoothing. The smooth operation is designed to reduce the difficulty of quantizing activations, while the UV-PTQ method uses a UV weight map to guide the quantization process. Together, these techniques enable efficient and accurate quantization of the Codec Avatar decoder for real-time inference on VR headsets.

from activations to weights through a mathematically equivalent transformation in transformers models. For a linear layer $Y = XW = (X \mathrm{diag}(s)^{-1}) \cdot (\mathrm{diag}(s)W)$ it introduces scaling factors $s$ to smooth activations.

We adapt this principle to transposed convolutions. For a ConvTranspose layer with input tensor $X \in \mathbb{R}^{C_{in} \times H_{in} \times W_{in}}$ and weight $W \in \mathbb{R}^{C_{in} \times C_{out} \times K_h \times K_w}$, let $s = (s_1, s_2, \ldots, s_{C_{in}})$ be a set of positive smooth factors, one for each input channel. Specifically, let $\tilde{X}$ denote the scaled input, where each channel $c$ is multiplied by its corresponding scale factor $s_c$. In other words, $\tilde{X}[c,:,:] = s_c \cdot X[c,:,:]$ for every channel $c \in C_{in}$, effectively scaling the activations of each channel by $s_c$. To preserve the output activations, define $\tilde{W}$ as the adjusted weight tensor for this layer, where the filter corresponding to input channel $c$ is scaled by $1/s_c$. Formally, for each input channel index $c$, $\tilde{W}[c,:,:,:] = \frac{1}{s_c} W[c,:,:,:]$. The proof of the equivalence between $\tilde{Y} = \tilde{X} * \tilde{W}$ and $Y = X * W$ is provided in Appendix A, where $*$ denotes the convolution operation. This formulation introduces a pair of transformations $s_c$ and $\frac{1}{s_c}$ for each channel. These transformations offset each other with respect to the layer output. The scale values are carefully determined using the calibration dataset to yield activations that are more suitable for quantization, consistent with observations from prior post-training smoothing methods [52, 3, 22, 23].

Importantly, ICAS incurs no runtime overhead, as the scale $s$ and its inverse are precomputed and fused into the network parameters using offline operation. Specifically, for a transposed convolution followed by nonlinear functions (e.g. LeakyReLU [53]), we fuse the scales into adjacent layers to avoid explicit multiplication at inference. For example, consider two consecutive layers $L_i$ and $L_{i+1}$ with a non-linear activation function $\sigma$ between layers.

$$X^{(i+1)} = \sigma(W^{(i)} * X^{(i)} + B^{(i)}), \quad X^{(i+2)} = \sigma(W^{(i+1)} * X^{(i+1)} + B^{(i+1)}) \tag{2}$$

where $X^{(i)}$ is the input to $L_i$, $W^{(i)}$ and $B^{(i)}$ are its weights and bias, respectively. Specifically, we have the following observation:

$$\tilde{X}^{(i+2)} = s \otimes X^{(i+2)}[,:,:] = (s \otimes \sigma(W^{(i)}[:,:,:,:] * X^{(i)} + B^{(i)}) \tag{3}$$

where $\otimes$ represents the elementwise product. Continuing with the Codec Avatar model, we adopt the LeakyReLU activation function [53], defined as $\sigma(x) = \max(\alpha x, x)$, where $\alpha$ denotes the negative slope. For any channel $c$ and scaling factor $s_c > 0$, it follows that $\sigma(s_c \cdot X[c,:,:]) = s_c \cdot \sigma(X[c,:,:])$. Hence,

$$\tilde{X}^{(i+2)} = \sigma((s \otimes W^{(i)}[:,,:,:]) * X^{(i)} + s \otimes B^{(i)}) \tag{4}$$

where $s \otimes W^{(i)}[:,,:,:]$ denotes the fused weight of layer $L_i$. The detailed proof is provided in Appendix B. Based on this formulation, the scales $s_c$ can be incorporated into the weight $W^{(i)}$ of the preceding layer by scaling each output channel of $W^{(i)}$ and the corresponding bias $B^{(i)}$ with the associated factor. We enforce $s_c > 0$ for all channels $c$ to preserve the sign of activations under smoothing. This eliminates explicit scaling operations during inference, as the calibrated scales are pre-fused into the convolutional weights during the offline calibration stage.

Inspired by the migration strategy of SmoothQuant [52], we determine the smoothing factor for each channel by balancing the dynamic ranges of the original activations and corresponding weights.

$$s_c = \frac{(\max_{m,n}|X[c,m,n]|)^\alpha}{(\max_{c_o,k,h}|W[c,c_o,k,h]|)^{1-\alpha}} \tag{5}$$

The exponent $\alpha \in [0,1]$ serves as a migration-strength hyperparameter. In practice, we sweep over different values of $\alpha$ and select the optimal value of 0.8. Here, $m$ and $n$ denote the spatial dimensions of the activation $X$; $k$ and $h$ represent the kernel dimensions of the weight filters; and $c$ and $c_o$ correspond to the input and output channels of the activations and filters, respectively.

### 3.2 Facial-Feature-Aware Smoothing

While ICAS uniformly scales all channels, certain feature channels correspond to critical facial details that should be preserved. We propose Facial-Feature-Aware Smoothing (FFAS) as a targeted refinement to ICAS. In a Codec Avatar decoder, outputs are often mapped to a texture space representing the face. We leverage predefined facial region masks to identify which feature maps carry high-frequency details in those regions. Concretely, for each channel $c$ in a given layer $l$ with input feature map of size $H^l \times W^l$ in texture space, we compute the activation variance within the important facial regions. Let $R_c^l \subseteq \{1, ..., H^l\} \times \{1, ..., W^l\}$ denote the set of texture pixels belonging to a particular facial region of interest. We measure the pixel-wise variance of channel $c$ over that region,

$$\sigma_c^2(R_c^l) = \frac{1}{|R_c^l|} \sum_{(m,n) \in R_c^l} (X^l[c,m,n] - \mu_c(R_c^l))^2 \tag{6}$$

where $X^l[c,i,j]$ is the activation value of channel $c$ at spatial location $(i,j)$ and $\mu_c(R_c^l)$ is the mean activation over region $R_c^l$. A large $\sigma_c^2(R_c^l)$ indicates that channel $c$ exhibits significant variation within facial region. We rank all channels by $\sigma_c^2$ and identify the top-$k\%$ channels with highest variance in critical regions. FFAS exempts these top-$k\%$ channels from ICAS smoothing, $k$ is a hyperparameter. For channels in this set, we effectively set $s_c = 1$ so that their activations remain at full magnitude. The remaining channels still receive smooth facotr determined by ICAS. By skipping smoothing on the most detail-sensitive feature maps, FFAS preserves high-frequency facial details like eye wrinkles and lip creases that might otherwise be attenuated by ICAS. Notably, this selection is data-driven and region-specific. In our experiments, integrating FFAS on the top of ICAS framework effectively mitigated over-smoothing in critical facial regions such as the eyes and mouth. This approach preserved essential expression details, reduced global artifacts, and enhanced the overall visual quality of the generated avatars.

### 3.3 UV-Weighted Post-Training Quantization

To preserve perceptually critical facial details under low-bit weight quantization, we propose a UV-mapping Weighted Post-Training Quantization (UV-Weighted PTQ) that uses view-dependent UV coordinates to guide a mask-weighted error minimisation. In computer graphics, UV coordinates denotes the 2-dimension texture domain onto which a 3-D surface mesh is parametrically unwrapped. Each vertex of the face mesh stores fixed $(u,v)$ coordinates, allowing the decoder to output a flat texture map that is later sampled during rendering. Consequently, pixels in UV regions corresponding to salient facial areas (eyes, mouth, nose) are perceptually critical, whereas others may be invisible or less important in the final view [10].

We compute per-feature UV coordinates by leveraging the existing rendering pipeline of the pretrained VAE decoder. The decoder outputs a 3D mesh $\hat{M} \in \mathbb{R}^{V \times 3}$ where $V$ is the number of vertices in the mesh. We then rasterize $\hat{M}$ onto the 2-D grid of feature map with $H \times W$ to obtain barycentric coordinates for each grid location. Assume $\phi_{u,p}$ and $\phi_{v,n}$ are the UV coordinates of pixel $p$, weight $w_p = \text{rasterize}(\hat{M}, H \times W)$, $[\phi_{u,p}, \phi_{v,p}] = \frac{\sum w_p \Phi_p}{\sum w_p}$. Using these weights, we interpolate the per vertex UV coordinates $\Phi_p \in [0,1]^2$ to each pixel. Then, we map the normalized coordinates to integer texel indices on the 2-D texture map of resolution $H \times W$, and accumulate a hit-count map $A \in \mathbb{N}^{H \times W}$. We normalize $A$, apply a upper bound $w_{max}$ and broadcast across channels to form the

soft importance weight $W_{uv} \in [0, w_{max}]^{C_{in} \times H \times W}$.

$$W_{uv}[:, m, n] = \begin{cases} \frac{A[m,n]}{max_{p,q}(A[p,q])} w_{max} & \text{if } A[m,n] \neq 0 \\ 0 & \text{if } A[m,n] = 0 \end{cases} \tag{7}$$

where $A[m,n] = \sum_{p=1}^{V} \mathbb{1}((\lfloor \phi_{u,p} H \rfloor, \lfloor \phi_{v,p} W \rfloor) = (m, n))$, $\mathbb{1}(\cdot)$ is the indicator function and $\lfloor \cdot \rfloor$ is the round function. For each layer $l$ in the decoder, we downsample UV weight to match layer's input size $H_l \times W_l$.

During quantization, our goal is to minimize the quantization error between the original and quantized weights. To achieve this, we utilize approximate second-order information by calibrating the weights using the Hessian matrix. When computing this matrix, the activations $X^l$ are pre-multiplied by the downsampled UV importance weights $W_{uv}^l$.

$$H_{uv}^l = \frac{1}{S} \sum_{s=1}^{S} (2(W_{uv}^l \cdot X_s^l) * (W_{uv}^l \cdot X_s^l)^T) + \lambda I \tag{8}$$

where $S$ is the number of samples, $X_s^l$ is the $s$-th sample of the input to layer $l$, and $\lambda$ is a small regularization term. Given the weighted Hessian $H_{uv}^l$, we follow the GPTQ [12] greedy quantization procedure. For each layer $l$ with unfold weight $W^l \in \mathbb{R}^{(C_{in} K_h K_w) \times C_{out}}$, we quantize weights column-by-column. For $r$-th column, we have

$$\hat{W}^l[:, r] = \text{quant}(W^l[:, r]), \quad e_r = W^l[:, r] - \hat{W}^l[:, r] \tag{9}$$

where quant$(\cdot)$ maps weights to the nearest quantized value. The quantization error $e_r$ is then compensated across remaining unquantized columns using the inverse Hessian.

$$W^l[:, j] \leftarrow W^l[:, j] - e_r \frac{H_{uv}^l[r, j]}{H_{uv}^l[r, r]}, \quad \forall j > r \tag{10}$$

The UV-weighting in $H_{uv}^l$ ensures that errors affecting critical facial regions are penalized heavily.

### 3.4 Input-Combining Mechanism for PCA Accelerator and Optimized Pipeline

As discussed in Section 1, VR head-mounted displays (HMDs) are typically resource-constrained, making the execution of PCA computationally expensive and slow. Moreover, performing PCA on the GPU or NPU within the HMD can heavily consume hardware resources and degrade the performance of other concurrently running VR applications. To address this issue, we design a dedicated hardware accelerator for efficient PCA execution. As illustrated in Figure 5 (a), the core of the accelerator is a $16 \times 16$ systolic array [19], a dataflow architecture consisting of a grid of interconnected processing elements (PEs). Each PE performs multiply–accumulate (MAC) operations and transmits intermediate results to neighboring PEs in a rhythmic, wave-like manner. This structure is highly efficient for matrix multiplication tasks. In our design, we employ a weight-stationary systolic array configuration, where weights are preloaded into the array and remain fixed during computation, while activations are streamed in from bottom to top in a staggered sequence. Partial sums flow horizontally from left to right and are collected from the rightmost column of PEs.

As detailed in Section 2.3, the transposed convolution and im2col transformation introduce extreme sparsity into the activation maps, leading to severe underutilization of the hardware accelerator. To mitigate this inefficiency, following the prior work [20], we propose an optimization technique called *input-combining* to compress the activation input and enhance hardware utilization. As shown in Figure 5 (b), the activation map is first partitioned into $4 \times 4$ tiles. These tiles are then categorized into two types: (1) tiles with checkerboard-like sparsity patterns; and (2) tiles that are entirely zero. We discard the fully zero tiles and vertically stack the remaining tiles to form a compact representation. This eliminates a significant portion of zero activations without loss of useful information.

To implement this combined input format, we modify the PE design as illustrated in Figure 5 (c). Every PE preloads two weights and accepts two activations per cycle, one of which would be zero. Two multiplexers are used to select the non-zero activation and its corresponding weight, allowing each PE to perform a single MAC operation per cycle. This lightweight enhancement enables the accelerator to bypass most zero activations, focusing computation only on non-zero data with minimal

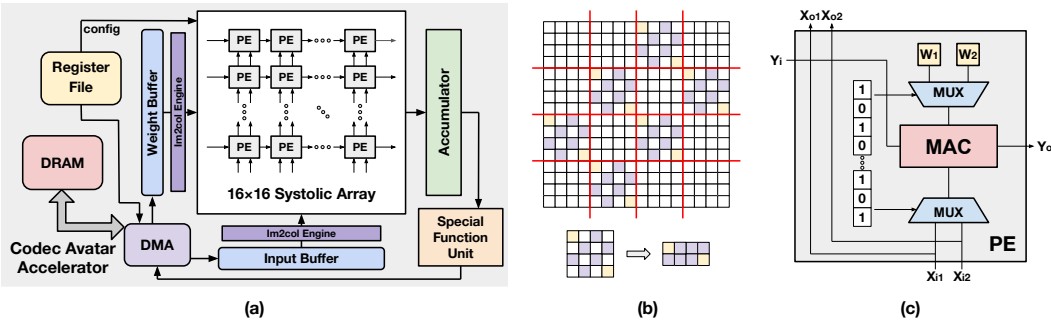

Figure 5: (a) Architecture of the proposed hardware accelerator for Codec Avatar inference. (b) Input-combining tiling scheme applied to the activation matrix. Red lines partition the input activation into smaller tiles. Purple/white squares denote non-zero/zero activations, and yellow squares are zero but can be assumed non-zero for simplicity. (c) Internal architecture of the proposed PE.

hardware overhead. In the ideal case, this strategy can reduce the number of operations by up to 75%, delivering significant latency improvements, as shown in the experiment results in Section 4.5.

As described in Section 2.1 and Figure 3 (a), the complete pipeline of Codec Avatar consists of five stages. To further reduce end-to-end latency, we propose an optimized execution pipeline. As illustrated in Figure 6 (b), Transmission and Decoding are performed in parallel, and multiple frames are processed in an overlapped fashion. Two scheduling constraints must be satisfied: (1) Decoding can only start after Encoding finishes, since both are executed on the same hardware accelerator; (2) Decoding must wait until the corresponding Transmission completes to receive the latent code from the remote user. This pipeline design fully exploits inter-frame parallelism and significantly improves overall throughput. The resulting end-to-end latency and frame rate are analyzed in Section 4.5.

## 4 Experiments

### 4.1 Experimental Setup

We extensively evaluate the proposed quantization methods on PCA decoding, focusing on visual quality and system performance under 4-bit and 8-bit settings. We evaluate our quantization method on the MultiFace dataset [50]. This dataset provides high-quality captures 65 scripted facial expressions, along with ground-truth textured 3D face meshes. For each expression, we use the provided 3D geometry and texture map ($1024 \times 1024$ UV atlas) to render the avatar from three camera viewpoints: one front and two approximately $45°$ side views.

Classical full-reference metrics such as Peak Signal-to-Noise Ratio (PSNR) [45] and Structural Similarity (SSIM) [49] measure absolute pixel-wise or local structural differences. Hence, they correlate weakly with human perception when small mis-alignments or high-frequency phase shifts are present, both of which are common in generative avatars [33]. We report the FovVideoVDP (VDP) metric [31], a full-reference perceptual metric that accounts for spatio-temporal human visual sensitivity designed for wide field-of-view VR content. We also report LPIPS [56] metric which compares feature activations of a pretrained network. All methods are evaluated on an NVIDIA A100 GPU and the hyperparameter $k = 75$ in our experiment.

As for model inference, we choose Snapdragon XR2 Gen 2 SoC as the baseline against our proposed hardware accelerator, because it is integrated within Meta Quest 3 headset, representing real-world deployment constraints. And we perform the rendering process on NVIDIA Tesla T4 16GB GPU (1590 Mhz clock, 2560 CUDA cores). The experiments are conducted under certain conditions to mimic the performance of an edge device GPU, specifically NVIDIA Jetson Orin NX 16GB [37] (918 MHz clock, 1024 CUDA cores), a mobile platform which has been frequently used in prior research to model rendering latency in VR headsets [15, 17, 55].

### 4.2 Baseline Methods

We compare the proposed quantization approach with several state-of-the-art baselines. Full Codec Avator [50] is the original model with no quantization. AdaRound [35] is an adaptive weight

Table 1: VDP [31] and LPIPS [56] scores for different methods at 4-bit and 8-bit quantization. Gray cells indicate our proposed methods. Best results are in **bold**.

| Method | Precision | Front | | Left | | Right | |
|---|---|---|---|---|---|---|---|
| | | VDP↑ | LPIPS↓ | VDP↑ | LPIPS↓ | VDP↑ | LPIPS↓ |
| Full Model [50] | FP32 | 6.5364 | 0.21604 | 5.9480 | 0.21965 | 5.8625 | 0.20428 |
| Adaround [35]+LSQ[9] | W4A4 | 4.2531 | 0.22612 | 3.6143 | 0.24000 | 3.5606 | 0.22031 |
| POCA [33] | | 5.2310 | 0.22200 | 4.3838 | 0.23643 | 4.3457 | 0.21347 |
| 2DQuant [25] | | 5.2987 | 0.22186 | 4.3948 | 0.23243 | 4.3712 | 0.21209 |
| GPTQ [12] | | 5.4980 | 0.22048 | 4.5868 | 0.23085 | 4.5729 | 0.21256 |
| ICAS (Ours) | | 5.5901 | 0.21981 | 4.7317 | 0.22783 | 4.7536 | 0.21127 |
| UV-W (Ours) | | 5.7559 | 0.21778 | 4.8130 | 0.22699 | 4.8187 | 0.21062 |
| ICAS-UV (Ours) | | 5.6438 | 0.21941 | 4.9145 | 0.22650 | 4.9057 | 0.20840 |
| FFAS-UV (Ours) | | **5.8541** | **0.21746** | **4.9795** | **0.22649** | **4.9605** | **0.20719** |
| Adaround [35]+LSQ[9] | W8A8 | 6.2106 | 0.21667 | 5.5004 | 0.22135 | 5.4381 | 0.20601 |
| POCA [33] | | 6.4827 | 0.21612 | 5.8511 | 0.22048 | 5.7565 | **0.20408** |
| 2DQuant [25] | | 6.4983 | 0.21645 | 5.8313 | 0.22088 | 5.7497 | 0.20436 |
| GPTQ [12] | | 6.2359 | 0.21687 | 5.6188 | 0.22101 | 5.3613 | 0.20546 |
| ICAS (Ours) | | 5.6007 | 0.21748 | 5.3913 | 0.22973 | 5.0762 | 0.20840 |
| UV-W (Ours) | | 6.5271 | 0.21610 | **5.9101** | **0.22036** | 5.7610 | 0.20543 |
| ICAS-UV (Ours) | | 6.3690 | 0.21682 | 5.6615 | 0.22091 | 5.5989 | 0.20541 |
| FFAS-UV (Ours) | | **6.5241** | **0.21605** | 5.8589 | 0.22068 | **5.8071** | 0.20441 |

rounding technique for post-training quantization. LSQ (Learned step size quantization)[9] is a method that learns the quantization scaling size for each layer during training. POCA (Post-training Quantization with Temporal Alignment) is a recent method tailored for codec avatars [33]. We also adapt 2DQuant [25], a two-stage PTQ method originally proposed for 4-bit image super-resolution models, to our avatar decoder. GPTQ [12] is a Hessian-based quantization method that uses layer-wise Hessian information to optimize weight updates. We also include a UV-only quantization method that applies quantization without any smoothing(UV-W), a smooth-only method that only applies Channel-wise Activation Smoothing without UV guidance (ICAS) and a smooth-UV method without Facial-Feature-Aware Smoothing (ICAS-UV).

Each baseline is applied to our pre-trained avatar decoder, and we evaluate both 8-bit (INT8) and 4-bit (INT4) quantization settings for all methods. For fair comparison, all models including baselines and our use the same pre-trained float32 decoder as a starting point and are calibrated on a small sample set of avatar frames (512 frames). VDP provide a more faithful estimate of user-perceived quality than PSNR/SSIM, and thus form the primary metrics in our study.

### 4.3 Low-bit Quantization Results

Table 1 demonstrates that our complete method achieves superior perceptual quality across all three camera perspectives at 4-bit quantization, surpassing the best-performing baseline (GPTQ) with improvements of +0.36/+0.39/+0.39 VDP for front/left/right views, respectively. These gains confirm that combining channel smoothing, UV-guided calibration, and facial-feature protection is crucial for perceptual realism. Ablations highlight complementary benefits: ICAS suppresses bursty channels, UV-W reallocates error to less-visible texels, and FFAS preserves fine eye-mouth details, together yielding the observed jump in temporal fidelity.

At 8-bit precision all methods converge to high quality, yet our methods still edges out the best baseline by up to 0.06 VDP. The ICAS-only variants underperform compared to other methods at 8-bit precision. This suggests that excessive smoothing can degrade important high-frequency details, leading to a reduction in VDP by 0.9 on the frontal view. When sufficient quantization levels are available, aggressive rescaling is unnecessary and may even be detrimental to perceptual quality. Moreover, our proposed method significantly reduces the quality gap between frontal and side views The performance gap from front to left is 0.88 for ours vs. 1.21 for GPTQ, indicating improved view-consistency. This property is particularly important for immersive VR applications, where users frequently change their gaze direction and head pose.

Table 2: FovVideoVDP scores for spontaneous facial expressions at 4-bit quantization (W4A4).

| Method | Shushing | | | Surprise | | | Frowning | | |
|---|---|---|---|---|---|---|---|---|---|
| | Front | Left | Right | Front | Left | Right | Front | Left | Right |
| GPTQ [12] | 5.2911 | 4.4226 | 4.3828 | 5.4066 | 4.5072 | 4.4783 | 5.2713 | 4.4064 | 4.3877 |
| FFAS-UV (Ours) | **5.7832** | **5.0065** | **5.0280** | **5.8486** | **5.0930** | **5.0989** | **5.8686** | **5.0210** | **5.0606** |

Figure 6: (a) Inference latency of PCA on different hardware platforms. (b) Optimized PCA pipeline.

## 4.4 Spontaneous Facial Expression Results

To assess how well ESCA generalizes to spontaneous facial expressions, we evaluate three diverse expressions from the MultiFace dataset's 65 expressions: shushing, surprise, and frowning. These expressions represent a range of facial dynamics including subtle mouth movements, wide-eye surprise, and brow furrowing. Table 2 presents the FovVideoVDP scores computed against ground truth across three viewpoints (front, left, and right) at 4-bit quantization. Our FFAS-UV method consistently outperforms the best baseline (GPTQ) across all expressions and viewpoints, with improvements ranging from $+0.49$ to $+0.66$ VDP. The results demonstrate that ESCA maintains high visual fidelity for spontaneous expressions while preserving consistent performance across viewing angles, confirming its robustness for real-world avatar applications.

## 4.5 Latency Results

We evaluate the latency improvement achieved by the proposed input-combining accelerator. The results are shown in Figure 6 (a). For the encoder, which consists solely of standard convolution layers, the inference latency is 3.05 ms under INT8 quantization and remains unchanged for baseline and input-combining accelerators. For the decoder, which is dominated by transposed convolution, the INT8 baseline accelerator achieves a latency of 42.04 ms, while the input-combining design reduces this to 12.51 ms, representing a $3.36\times$ speedup. Under INT4 quantization, our input-combining accelerator achieves a minimum latency of 3.13 ms.

We adopt latency references from prior VR research: camera sensor acquisition takes approximately 1 ms [2, 24, 44], while Wi-Fi 6 transmission requires around 5 ms under favorable conditions [57]. Our accelerator executes both Encoding and Decoding in approximately 3 ms each. And Rendering on GPU requires 9.5 ms under the configuration in Section 4.1. With the optimized pipeline introduced in Section 3.4, the effective per-frame latency, indicated by the interval between the two red dashed lines in Figure 6 (b), is determined by twice the Transmission delay, totaling 10 ms. Consequently, the effective frame rate reaches $1000/10 = 100$ FPS, achieving significant throughput improvement and fully satisfying the real-time requirement for immersive Codec Avatar applications.

## 5 Conclusion

We have presented ESCA, a comprehensive framework that co-optimizes neural network algorithms and hardware design to enable real-time PCA inference on VR devices. ESCA combines two smoothing techniques (ICAS and FFAS) with a UV-weighted Hessian-based quantization strategy to achieve high fidelity at 4-bit precision, and it includes a custom accelerator optimized for transposed convolutions that yields a $3.36\times$ reduction in decoder latency. Despite these promising results, several limitations remain. It relies on accurate facial UV priors and only accelerates the decoding stage, leaving other parts of the pipeline unoptimized. Future work will aim to reduce dependence on UV maps and extend co-optimization to the full avatar pipeline for more seamless systems.

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

## A  Proof of Scaling Invariance

Let the convolution layer take an input tensor $X \in \mathbb{R}^{C_{\text{in}} \times H \times W}$ and a weight tensor $W \in \mathbb{R}^{C_{\text{out}} \times C_{\text{in}} \times k_h \times k_w}$. For clarity we fix a single output channel and suppress the output-channel index; the argument is identical for every output filter. With the usual definition of discrete convolution ($*$), the pre-activation output is

$$Y = \sum_{c=1}^{C_{\text{in}}} W[c] * X[c], \tag{11}$$

where $W[c] \in \mathbb{R}^{k_h \times k_w}$ and $X[c] \in \mathbb{R}^{H \times W}$ denote the $c$-th input-channel kernel and feature map, respectively.

Choose positive scalars $s_1, \dots, s_{C_{\text{in}}}$. Define the scaled activations and the compensated weights

$$\tilde{X}[c] = s_c \, X[c], \qquad \tilde{W}[c] = \frac{1}{s_c} \, W[c], \qquad c = 1, \dots, C_{\text{in}} \tag{12}$$

For any scalar $\alpha \in \mathbb{R}$ and tensors $A, B$ of compatible shape, convolution is bilinear:

$$(\alpha A) * B = \alpha \, (A * B), \quad A * (\alpha B) = \alpha \, (A * B) \tag{13}$$

Using Equation 13 with $\alpha = s_c$ and $\alpha = 1/s_c$,

$$(\tfrac{1}{s_c} W[c]) * (s_c X[c]) = \tfrac{1}{s_c} \, s_c \, (W[c] * X[c]) = W[c] * X[c] \tag{14}$$

Summing Equation 14 over all input channels reproduces Equation 11:

$$\tilde{Y} = \sum_{c=1}^{C_{\text{in}}} \tilde{W}[c] * \tilde{X}[c] = \sum_{c=1}^{C_{\text{in}}} W[c] * X[c] = Y \tag{15}$$

Channel-wise scaling of activations, paired with the reciprocal scaling of the corresponding kernels, leaves the convolution output unchanged. Hence $\tilde{Y} = Y$, proving the scale invariance claim.

## B  Proof of Fusing Scaling into Previous Layer Weights

Let

$$X^{(1)} \in \mathbb{R}^{C_{\text{in}} \times H \times W}, \quad W^{(1)} \in \mathbb{R}^{C_{\text{out}} \times C_{\text{in}} \times k_h \times k_w}, \quad B^{(1)} \in \mathbb{R}^{C_{\text{out}}} \tag{16}$$

and define the

$$X^{(2)} = W^{(1)} * X^{(1)} + B^{(1)} \tag{17}$$

Let $s \in \mathbb{R}_{>0}^{C_{\text{out}}}$ be a per-output-channel scale, and write $s \otimes T$ for broadcast Hadamard multiplication along all trailing dimensions of a tensor $T$ whose first index has size $C_{\text{out}}$.

**Claim.**
$$\widetilde{X}^{(2)} = s \otimes X^{(2)}[,:,:] = (s \otimes W^{(1)}[:,,:,:]) * X^{(1)} + s \otimes B^{(1)} \tag{18}$$

Fix an output channel $c \in \{1, \dots, C_{\text{out}}\}$ and spatial index $(i, j)$. By definition of convolution with bias,

$$X_{c,i,j}^{(2)} = \sum_{m=1}^{C_{\text{in}}} \sum_{u=0}^{k_h-1} \sum_{v=0}^{k_w-1} W_{c,m,u,v}^{(1)} X_{m,i-u,j-v}^{(1)} + B_c^{(1)} \tag{19}$$

Multiply Equation 19 by the scalar $s_c$:

$$s_c \, X_{c,i,j}^{(2)} = \sum_{m,u,v} \left( s_c \, W_{c,m,u,v}^{(1)} \right) X_{m,i-u,j-v}^{(1)} + s_c \, B_c^{(1)} \tag{20}$$

The summation term in Equation 20 is exactly the $(c, i, j)$-entry of the convolution $(s \otimes W^{(1)}) * X^{(1)}$, while the final term is the $(c, i, j)$-entry of $s \otimes B^{(1)}$ (broadcast spatially). Since (2) holds for every $c, i, j$, we obtain

$$s \otimes X^{(2)} = (s \otimes W^{(1)}) * X^{(1)} + s \otimes B^{(1)} \tag{21}$$

which proves the claim.

Scaling each output channel by $s$ can be equivalently implemented by scaling the corresponding output-channel kernels in $W^{(1)}$ before convolution. In quantization or inference-time fusion, this lets us absorb channel-wise activation rescaling into the layer's weights, avoiding an extra runtime operation.

## C  Proof of scaling invariance before and after im2col

Let $X \in \mathbb{R}^{C_{in} \times H_{in} \times W_{in}}$ be the activation tensor feeding a ConvTranspose2d layer. $\text{im2col}(\cdot)$ convert its argument to a 2-D matrix in which every column is the receptive-field patch that contributes to one output location.

$$X_{\text{col}} = \text{im2col}(X) \in \mathbb{R}^{(C_{in} K_h K_w) \times N} \tag{22}$$

where $N = H_{\text{out}} W_{\text{out}}$ is the number of spatial sites produced by the layer.

$W \in \mathbb{R}^{C_{in} \times C_{out} \times K_h \times K_w}$ be the kernel of the transposed convolution, reshaped into

$$W_{mat} = \text{reshape}(W) \in \mathbb{R}^{(C_{in} K_h K_w) \times (C_{out})} \tag{23}$$

After im2col, transposed convolution is a plain matrix multiply followed by col2im accumulation

$$Y_{col} = W_{mat}^T X_{col} \tag{24}$$

then $Y = \text{col2im}(Y_{col})$.

Let the per-channel scale vector be $s = [s_1, \ldots, s_{C_{in}}]^\top$ with $s_c > 0$. Define $S_{act} = diag(s) \otimes I_{K_h K_w}$, where $I_{K_h K_w}$ is the identity matrix of size $K_h \times K_w$. The Kronecker product $\otimes$ constructs a block-diagonal matrix $S_{act}$, and $S_{act} \in \mathbb{R}^{(C_{in} K_h K_w) \times (C_{in} K_h K_w)}$. Every column block belonging to channel $c$ is multiplied by the same scalar $s_c$.

We scale activations and invert-scale the weights $\tilde{X_{col}} = S_{act} X_{col}$ and $\tilde{W_{mat}} = S_{act}^{-1} W_{mat}$.

Propagating the scaled quantities through the same algebra as (24).

$$\tilde{Y_{col}} = \tilde{W}_{mat}^\top \tilde{X}_{col} = (S_{act}^{-1} W_{mat})^\top (S_{act} X_{col}) \tag{25}$$

From $S_{act}$ is diagonal, $S_{act}^{-1} = S_{act}^{-1T}$.

$$\tilde{Y}_{col} = W_{mat}^\top S_{act}^{-1} S_{act} X_{col} = W_{mat}^\top X_{col} = Y \tag{26}$$

