# OpenReview forum: "ESCA: Enabling Seamless Codec Avatar Execution through Algorithm and Hardware Co-Optimization for Virtual Reality"
_NeurIPS.cc/2025/Conference — NeurIPS 2025 poster_

### Official Review · Reviewer_CJWi · 2025-06-29

**Clarity:** 3
**Significance:** 4
**Originality:** 4
**Rating:** 6
**Confidence:** 4

**Summary:**

The paper presents ESCA, a framework for accelerating Code Avatar inference on AR/VR devices. Authors concentrate on two challenges: preserving fidelity during 4-bit quantization of VAE  and inefficient inference of transposed convolutions on target devices. They propose a number of techniques, including two smoothing techniques, uv-weightd post-training quantization and designing a custom accelerator. The resulting pipeline runs under impressive 100 fps while preserving and even improving a bit the visual quality.

**Questions:**

1. what is offline time and memory footprint of the techniques you use
2. how well does it perform on spontaneous facial exprresions?

**Ethical Concerns:**

["NO or VERY MINOR ethics concerns only"]

**Final Justification:**

After reading the rebuttal, I raise my score. The authors show their smoothing does not change the convolution output and keeps key facial details; the eye/mouth metrics improve over baselines. The quantization is post-training and light with clear memory figures. The system reaches ~100 fps on device and keeps quality, including for spontaneous expressions.

**Limitations:**

yes

**Paper Formatting Concerns:**

no concerns

**Quality:**

4

**Strengths And Weaknesses:**

Strenghts:
* achieves real-time 100 fps on device
* new accelerator design that exploits structures sparsity
* paper addresses both algorithmic and hardware problems
* clear system diagrams, demonstrating the whole pipeline picture


Weaknesses:
* Little qualitative results. For example, it is especially intresting how facial feature-aware smoothing actually affects the visual quality of the important features (eyes, mouth) compared to others
* limited discussion of training cost and calibration overhead for ICA/UV-PTQ
* evaluation is limited to MultiFace dataset

---

> ### Author Rebuttal · Authors · 2025-07-30
>
> Thank you for your insightful comments. We have summarized your questions below and provided detailed responses to each.
>
> ### Weaknesses and Questions
>  ### Q1. Little qualitative results. For example, it is especially interesting how facial feature-aware smoothing actually affects the visual quality of the important features (eyes, mouth) compared to others.
> **Ans:** Smoothing alone does not impact visual quality, as mathematically proven in our paper. It significantly reduces quantization noise by removing outliers, yet the process is effectively lossless—it does not alter the output of the convolution operation. Specifically, FFAS preserves critical facial details by identifying channels responsible for high-frequency features (eye wrinkles, lip creases) and exempting them from aggressive smoothing (Sec 3.2, lines 209-213). We analyzed reconstruction quality specifically within eye and mouth regions using facial masks. Results show FFAS-UV significantly outperforms both our ablation (ICAS-UV) and baseline methods:
>
> | Method             | Front PSNR | Front SSIM | Left PSNR | Left SSIM | Right PSNR | Right SSIM |
> |--------------------|------------|------------|-----------|-----------|------------|------------|
> | **FFAS‑UV (Ours)** | 35.501 | 0.920   | 34.255 | 0.919  | 34.040  | 0.920   |
> | **ICAS‑UV (w/o FFAS)** | 34.040 | 0.919 | 32.739 | 0.917  | 32.200021  | 0.918   |
> | **Baseline**       | 32.732  | 0.904   | 31.907 | 0.900  | 31.244  | 0.900   |
>
> This demonstrates FFAS successfully preserves visual quality in the most perceptually important facial features. We'll add more qualitative results in the next version due to rebuttal figure restrictions.
>
> ### Q2. Limited discussion of training cost and calibration overhead for ICA/UV-PTQ
>
> **Ans:** We would like to emphasize a key aspect of our framework: it is entirely post-training quantization (PTQ) and requires no expensive gradient-based training or backpropagation. The overhead is a one-time, offline calibration process. The calibration process is highly efficient. For each layer in the decoder, the overhead on an NVIDIA A100 GPU is as follows, using a calibration set of 512 avatar frames:
>
> | Stage                   | Description                                                 | Approx. Time per Layer     |
> |------------------------|-------------------------------------------------------------|----------------------------|
> | UV‑W Mask Generation   | Generate UV‑weighted importance mask from 512 samples.      | ~ 30 seconds (one‑time cost) |
> | ICAS Calibration       | Single forward pass over 512 samples (batch size 16) to find smoothing factors. | ~ 3 minutes               |
> | Weight Quantization    | Apply UV‑weighted Hessian and solve for the optimal quantized weights. | < 1 minute                 |
> | Total Calibration Cost | Offline, one‑time process                                     | ~ 4 minutes per layer      |
>
> We will include a detailed breakdown of these calibration costs in the Appendix of the next version to enhance clarity. More discussion is at Question 4.
>
> ### Q3. Evaluation is limited to MultiFace dataset
>
> **Ans:** To the best of our knowledge, MultiFace is the only public Codec-Avatar dataset with pretrained models, enabling post-training quantization experiments. Alternative datasets (Goliath [1], Tele-ALOHA [2]) provide only raw data requiring extensive training unfeasible within rebuttal timeframes [11].
>
> Despite single dataset use, MultiFace provides rigorous evaluation: 65TB with 13,000+ 2K frames per subject across 65 expressions and multiple viewpoints. The comprehensive coverage of extreme poses, occlusions, and high-frequency details creates a demanding testbed validating low-precision decoder performance.
>
> We plan broader evaluation once additional pretrained models become available. Current MultiFace evaluation represents the most scientifically sound approach given practical constraints while comprehensively validating quantization robustness.
>
> Although we did not evaluate ESCA on other codec avatar tasks, we show the applicability of ESCA to image super-resolution tasks, whose goal is to increase the resolution of an image. Although our UV-weighted PTQ cannot be extended to this domain due to the absence of UV coordinates, our smoothing can be used to facilitate the quantization over these super-resolution tasks.
>
>
> We compare our method against GPTQ, the best-performing baseline from Codec-Avatar quantization, across five commonly used super-resolution benchmarks [3, 4, 5, 6, 7]. Our quantization experiments used the SwinIR [8] super-resolution model, which was pretrained on the combined DIV2K [9] and Flickr2K [10] datasets.
>
> | Method   | Bit | Set5 PSNR | Set5 SSIM | Set14 PSNR | Set14 SSIM | B100 PSNR | B100 SSIM | Urban100 PSNR | Urban100 SSIM | Manga109 PSNR | Manga109 SSIM |
> |----------|-----|-----------|-----------|------------|------------|-----------|-----------|----------------|----------------|----------------|----------------|
> | SwinIR R | 32  | 31.13 dB  | 0.8967    | 27.45 dB   | 0.7867     | 26.37 dB  | 0.7434    | 25.15 dB       | 0.7993         | 29.60 dB       | 0.9115         |
> | ESCA     | 4   | 28.63 dB  | 0.8310    | 25.85 dB   | 0.7277     | 25.29 dB  | 0.6885    | 22.87 dB       | 0.6852         | 26.79 dB       | 0.8088         |
> | GPTQ     | 4   | 27.90 dB  | 0.7729    | 25.41 dB   | 0.6776     | 24.74 dB  | 0.6338    | 22.68 dB       | 0.6462         | 25.94 dB       | 0.7555         |
>
> ESCA achieves superior 4-bit quantization performance compared to GPTQ across all five super-resolution benchmarks, with PSNR improvements ranging from 0.19 dB (Urban100) to 0.85 dB (Manga109) and SSIM improvements ranging from 0.039 (Urban100) to 0.0581 (Set5).
>
> [1] Codec avatar studio: Paired human captures for complete, driveable, and generalizable avatars
>
> [2]  Tele-Aloha: a low-budget and high-authenticity telepresence system using sparse RGB cameras
>
> [3] Low-complexity single-image super-resolution based on nonnegative neighbor embedding
>
> [4] On single image scale-up using sparse-representations
>
> [5] A database of human segmented natural images and its application to evaluating segmentation algorithms and measuring ecological statistics
>
> [6] Single image super-resolution from transformed self-exemplars
>
> [7] Sketch-based manga retrieval using manga109 dataset
>
> [8] Swinir: Image restoration using swin transformer
>
> [9] Ntire 2017 challenge on single image super-resolution: Methods and results
>
> [10] Enhanced deep residual networks for single image super-resolution
>
> [11] High-quality joint image and video tokenization with causal VAE
>
> ### Q4. What is the offline time and memory footprint of the techniques you use?
>
> **Ans:** ESCA adopts a post-training quantization (PTQ) approach, which involves a one-time, offline calibration process that is significantly faster and more resource-efficient than full Quantization-Aware Training (QAT). To provide a clear summary, we have detailed the approximate calibration time and peak memory usage for our model, conducted on a single NVIDIA A100 GPU.
> | Resource Metric            | Description                                                              | Approximate Cost           |
> |---------------------------|--------------------------------------------------------------------------|----------------------------|
> | Calibration Time (per layer) | Time to calibrate one layer (ICAS + UV‑W PTQ) using 512 samples.       | < 4 minutes per layer      |
> | Calibration Time (total)     | Total one‑time, offline calibration time for the entire network.       | ~ 1 hour                   |
> | QAT Time                     | Total training time with A100×4 GPU setup.                             | > 1 day                    |
> | Calibration Memory Footprint | Peak GPU memory usage during the offline calibration process.           | ~ 60 GB                    |
> | QAT Memory Footprint         | Peak GPU memory usage during the QAT process.                          | ~ 200 GB                   |
>
> For the inference of our model, we have also detailed the estimated peak memory usage.
> | Model                 | Device                   | Source               | Peak Memory Usage |
> |-----------------------|--------------------------|----------------------|--------------------|
> | Encoder               | Snapdragon XR2 Gen 2     | Qualcomm AI Hub      | ~ 39 MB            |
> | Decoder               | Snapdragon XR2 Gen 2     | Qualcomm AI Hub      | ~ 50 MB            |
> | Encoder & Decoder     | Hardware Accelerator     | Hardware Simulation  | ~ 7 MB             |
>
> PTQ delivers 24x faster calibration with 70% less memory than QAT, while custom hardware achieves 87% memory reduction over mobile processors.
>
> ### Q5. How well does it perform on spontaneous facial expressions?
>
> **Ans:** We evaluated spontaneous facial expressions by selecting three expressions from the MultiFace dataset's 65 expressions and computing FovVideoVDP scores against ground truth across three viewpoints.
>
> | Method            | Shushing Front | Shushing Left | Shushing Right | Surprise Front | Surprise Left | Surprise Right | Frowning Front | Frowning Left | Frowning Right |
> |-------------------|----------------|---------------|----------------|----------------|----------------|-----------------|----------------|----------------|-----------------|
> | **FFAS‑UV**       | 5.7832         | 5.0065        | 5.0280         | 5.8486         | 5.0930         | 5.0989          | 5.8686         | 5.0210         | 5.0606          |
> | **Best Baseline** | 5.2911         | 4.4226        | 4.3828         | 5.4066         | 4.5072         | 4.4783          | 5.2713         | 4.4064         | 4.3877          |
>
> The results demonstrate ESCA maintains high visual fidelity for spontaneous expressions, with consistent performance across viewing angles.

---

### Official Review · Reviewer_52i2 · 2025-07-01

**Clarity:** 3
**Significance:** 3
**Originality:** 3
**Rating:** 4
**Confidence:** 3

**Summary:**

ESCA presents a full-stack framework for running photorealistic Codec Avatars in real-time on resource-constrained AR/VR devices. Majorly targeting on the VAE decoder side, it aims to addresse the quantization and efficiency challenges of transposed convolution-heavy decoders by co-designing algorithmic and hardware solutions. Key components include ICAS, FFAS, UV-weighted perceptual loss, and a custom DNN accelerator optimized for sparse patterns. ESCA achieves state-of-the-art 4-bit inference quality and enables 100 FPS rendering with 3.36× latency reduction.

**Questions:**

Current experiments and ablation studies are clear and effective in illustrating the contribution of each module to the performance gains.
Are there alternative design choices that could further accelerate each component? and possibly to discuss the theoretical upper bound of achievable latency reduction under the current framework.

Additionally, it would be helpful to understand potential directions for mitigating remaining artifacts, such as flickering in hair regions and loss of detail around the eyes.

How complex can the avatar subject be, e.g., can the system support full-body avatars with expressive dynamics? Providing more architectural details (e.g., latent space dimensionality, image resolution, and overall model size) would help better assessment.

**Ethical Concerns:**

["NO or VERY MINOR ethics concerns only"]

**Final Justification:**

After reading the rebuttal, I'll maintain my score.

**Limitations:**

As mentioned in previous weakness section, the scope of impact is of one major concern. Grounding each module design in broader conceptual principles, for example, by discussing how these components might migrate to emerging avatar decoder architectures with predictive capabilities would significantly enhance the generality of the work and help situate the proposed method within a wider design landscape. Expanding the discussion to include upstream and downstream components would provide readers a more complete picture of the system-level context.

**Quality:**

3

**Strengths And Weaknesses:**

The paper is clearly written and well-structured and easy to follow. The experimental evaluation is thorough. I can see the approach will directly addresses many practical deployment bottlenecks for high-fidelity codec avatar decoding in AR/VR, if current architecture is fixed.
Overall, ESCA presents a well-integrated co-design strategy for adapting quantization and accelerator design specifically to the decoder side of Codec Avatars - ICAS and DNN Hardware Accelator are reasonably adapted, while UV-weighted quantization and facial-feature-aware smoothing are novel in combining perceptual significance with quantization robustness.

While the proposed framework is well-optimized for Codec Avatar decoders, my main concern is that its scope appears tightly coupled to this specific architecture and application. Any substantial change in the decoder structure may necessitate rethinking or redesigning key components of the system. The demonstrated rendering results (in supp) still exhibit noticeable flickering artifacts (e.g., hair regions). This suggests a degree of reliance on human priors when selecting important regions and likely not generalize well to other subject types, e.g., even for full-body avatars, where semantic saliency and deformation patterns differ.
Experiments on larger-scale avart datasets would also be necessary.

Some figures (e.g., Figure 1, 2) are overly compressed within the text, making them slightly difficult to read.

---

> ### Author Rebuttal · Authors · 2025-07-30
>
> Thank you for your insightful comments. Below, we have summarized your questions and provided detailed responses to each one.
>
> ### Weaknesses and Questions
>
> ### Q1. While the proposed framework is well-optimized for Codec Avatar decoders, my main concern is that its scope appears tightly coupled to this specific architecture and application.
>
> **Ans:** We agree that ESCA is specialized by design, this is essential for addressing the challenging task of real-time photorealistic avatar inference on mobile AR/VR hardware. The concept of hardware-algorithm co-design has been widely adopted to support other AR/VR applications as well, such as image rendering [1–4], demonstrating its effectiveness in meeting stringent performance and efficiency requirements.
>
> Moreover, our core techniques have broader applicability. Input Channel-wise Activation Smoothing addresses activation outliers in generative models using transposed convolutions followed by non-linearities, which is a typical pattern in VAEs and GANs. Our channel-wise scaling and weight fusion method (Sec. 3.1, Eq. 2) is architecture-agnostic. Input-combining hardware optimization exploits the inherent checkerboard sparsity pattern from transposed convolutions after im2col transformation (Fig. 3a). This sparsity is a fundamental property of the operation itself, making our accelerator efficient for any decoder heavily using transposed convolutions for upsampling. Perceptually Guided Quantization (Sec. 3.2/3.3) is the most domain-specific component, designed to leverage facial UV priors and concentrate precision on perceptually critical regions for low-bit quantization.
>
> Similar to how ray tracing accelerators are optimized for specific traversal patterns [2] and 3D reconstruction systems codesign algorithms with hardware [1], codec avatars require specialized optimization due to their real-time requirements. General purpose solutions cannot achieve the 90+ FPS needed for immersive VR on resource constrained hardware.
> In general, ESCA offers valuable insights that can be applied to the optimization of other AI models across a range of tasks.
>
> [1] Fusion-3d: Integrated acceleration for instant 3d reconstruction and real-time rendering
>
> [2] CoopRT: Accelerating BVH Traversal for Ray Tracing via Cooperative Threads
>
> [3] AQB8: Energy-Efficient Ray Tracing Accelerator through Multi-Level Quantization
>
> [4] Neural rendering and its hardware acceleration: A review
>
> ### Q2. The demonstrated rendering results (in supp) still exhibit noticeable flickering artifacts. Experiments on larger-scale avatar datasets would also be necessary.
>
> **Ans:** We agree that our approach leverages semantic priors, which is an executive design choice to address the core challenge of 4-bit quantization for high-fidelity generative models. Under aggressive 4-bit quantization, perfect reconstruction everywhere is challenge. Our approach strategically allocates limited precision to perceptually critical facial regions (eyes, mouth) where users focus attention and convey emotion. This leverages psychovisual research showing viewer sensitivity to facial artifacts [5, 6].
>
> Regarding generalization: the principle of semantic-guided quantization is highly adaptable. For full-body avatars, the same methodology applies using full-body UV maps and prioritizing other salient regions (face, hands). Our work establishes this perceptually-guided PTQ framework for facial avatars, demonstrating a methodology extensible to other domains.
>
> Regarding dataset scale: MultiFace contains over 13,000 2k-resolution frames per subject across  65 diverse expressions, totaling 65TB, making it the largest available dataset with pretrained models. By comparison, FaceScape [7] supplies total 18760 textured meshes and requires approximately 350 GB of storage. 4DFAB [8] records 1.8 M high‑res meshes across repeated sessions and takes around 200 GB.
>
> Due to the characters restriction, more information is available at our response to Reviewer 6fss's Question 3 regarding additional evaluation concerns.
>
> [5] Influence of video distortions on facial expression recognition & quality
>
> [6] InternVQA: Advancing Compressed Video Quality Assessment with Distilling Large Foundation Model
>
> [7] Facescape: a large-scale high quality 3d face dataset and detailed riggable 3d face prediction
>
> [8] 4dfab: A large scale 4d database for facial expression analysis and biometric applications
>
> ### Q3. Are there alternative design choices that could further accelerate each component?
>
> **Ans:** Yes.  While our current framework represents a highly optimized design point, there are promising avenues for further acceleration, both at the algorithmic and hardware levels.
>
> For the Algorithm:
> One could explore more aggressive mixed-precision quantization, using our post components to automatically assign different bit widths to layers. Our quantization methods are largely orthogonal to and can be combined with other efficiency techniques like network pruning. One could first apply pruning and then apply our PTQ pipeline to the pruned model. While our current paper focuses specifically on advancing the state of the art in quantization for this domain, our method is designed to be a compatible component within a broader optimization toolkit.
>
> For the Hardware:
> Without any optimization, the latency is bottlenecked by the computation components of the framework (lines 108-111), namely, Encoding and Decoding. Therefore, our accelerator focuses on accelerating them where we use a systolic array, which is well known to be effective in operating matrix multiplication, and widely adopted in commercial AI accelerators such as Google’s TPU. Furthermore, we propose a novel input-combining mechanism to exploit the structured sparsity of the activation matrix, which significantly optimizes the performance of the accelerator.
>
> A further design can be applied on top of ours, because now we notice that after optimizing the computation, the bottleneck becomes the Transmission (lines 333-348). Therefore, further work can be done on minimizing the Transmission latency.
>
> ### Q4. The theoretical upper bound of achievable latency reduction.
>
> **Ans:** Our decoder acceleration reduces processing latency from 25.68 ms to 3.13 ms. However, as discussed in Section 4.4 (lines 333–348), the end-to-end frame latency of the Codec Avatar application is no longer constrained by computation but by Transmission latency. As shown in Figure 6(b), the effective per-frame latency, indicated by the interval between the two red dashed lines, is determined by twice the Transmission latency, totaling 10 ms. Without optimizing the Transmission latency, this will remain the upper bound for ESCA’s overall latency, regardless of further improvements in computation speed.
>
>
> ### Q5. Potential directions for mitigating remaining artifacts.
>
>
> **Ans:** Here are several promising future directions to build upon our work:
> Our current work uses uniform 4-bit or 8-bit quantization. A powerful extension would be to apply mixed precision, where different layers are assigned different bit-widths based on their sensitivity. For instance, layers responsible for fine facial features could maintain higher precision (8-bit) while background synthesis layers use aggressive 4-bit quantization, potentially reducing flickering in hair regions while preserving eye detail.
>
> ### Q6. How complex can the avatar subject be, e.g., can the system support full-body avatars with expressive dynamics? Providing more architectural detail.
>
> **Ans:** Our work targets facial avatars which is critical for social VR. The framework can extend to full-body avatars: UV-Weighted PTQ works with any UV texture map, and our accelerator optimizes transposed convolutions used across generative models.
> | Output texture resolution                  | 1024×1024×3 (Sec. 4.1, line 278) |
> |--------------------------------------------|----------------------------------|
> | Latent space              | 256×1                          |
> | Image resolution after rendering           | 2048×1334×3                    |
> | Encoder size                               | 6.79 M                         |
> | Decoder size                               | 24.68 M                        |
> | Number of transposed convolution layers    | 8                              |
> | Kernel size of transposed convolution layers | 4×4 (stride: 2, padding: 1)    |
> | Total model size                           | 31.47 M                        |
>
> We will add these details in the next version to improve clarity. The computational bottleneck is the decoder's transposed convolutions, not parameter count.
>
> ### Q7. The scope of impact is of one major concern. Expanding the discussion to include upstream and downstream components would provide readers a more complete picture of the system-level context.
> **Ans:**  Our method's design patterns can extend to emerging predictive decoders like Auto-CARD's LATEX framework [paper citation 12] and the dual-path architecture of ReliaAvatar [9], enabling integrated Codec Avatar pipelines.
>
> Upstream Integration: Our quantized decoder efficiently processes compressed representations from multiple sensing modalities (3D Gaussian splatting, NeRF, ray tracing, eye/gaze/hand tracking, long-context video understanding) while maintaining quality and reducing computational bottlenecks in sophisticated input processing pipelines.
>
> Downstream Benefits: Quantized latent representations preserve detail for high-quality rendering across diverse display technologies. By reducing decoder latency from 42.04ms to sub-10ms, we enable >90 FPS real-time rendering required for immersive VR while preventing frame drops and maintaining visual quality across display hardware.
>
> [9] Reliaavatar: A robust real-time avatar animator with integrated motion prediction
> ### Paper Format
> We will carefully adjust the layout and sizing of all figures, particularly Figures 1 and 2.

---

> > ### Comment · Reviewer_52i2 · 2025-08-06
> >
> > Thank you for the additional clarifications. The rebuttal has addressed most of my major concerns. I'll maintain my score.

---

> > > ### Author Response · Authors · 2025-08-08
> > >
> > > Thank you for your time and consideration throughout this process. We truly appreciate your insights and engagement.

---

### Official Review · Reviewer_p8Jg · 2025-07-03

**Clarity:** 2
**Significance:** 2
**Originality:** 3
**Rating:** 3
**Confidence:** 3

**Summary:**

This paper proposes a new hardware chip and several quantization methods tailored for encoder decoder blocks common among avatar codecs.

**Questions:**

1. Creating a new custom hardware is quite expensive, and it usually requires very strong motivations and business justifications. I think for proposing such a new hardware, the paper needs to first show the case for not being able to leverage existing support on AI accelerators for sparse operations. For example, similar sparse data pattern also show up in variants of attention heads. How does the proposed architecture compare with those baselines with all sparsity features leveraged?

 2. Are the latency results in Figure 6.a comparing Snapdragon XR2 Gen2 with NVIDIA A100? In general, lines 287-292 need further clarification about the choices in the benchmarking hardware and more explanation of the reasoning behind it.

3. What prevents your proposed hardware from further reducing the latency?

4. Do you have any evaluation or back of the envelope calculations re the energy consumption comparisons with baselines?

**Ethical Concerns:**

["NO or VERY MINOR ethics concerns only"]

**Limitations:**

Yes

**Paper Formatting Concerns:**

Please revise lines 287-292

**Quality:**

3

**Strengths And Weaknesses:**

The optimization techniques introduced for avatar generation are cleverly designed and their effectiveness is well-supported by informed evaluation metrics, successfully enabling 4-bit quantization. The problem that this paper is trying to address is timely and has established real-world applications.

However, the paper still has some primary weaknesses too:

    - Insufficient Justification for Custom Hardware: Proposing custom hardware is a significant undertaking that requires strong justification. The authors should first demonstrate why existing AI accelerators with support for sparse operations are inadequate. For instance, a comparison with baselines that leverage sparsity features is necessary to validate the need for a new architecture.

    - Unclear Benchmarking: The latency results presented in Figure 6a are ambiguous. It is not clear whether the comparison between a Snapdragon XR2 Gen 2 and the NVIDIA GPU is fair in terms of energy consumption and chip technology price point.

Additionally, a minor point is that the paper's title, "Algorithm and Hardware Co-Optimization," may be a misnomer. The contributions appear to be separate optimizations at the algorithm (quantization) and hardware levels, rather than a true co-optimization where the design of each is mutually influential. The proposed hardware does not seem specifically tailored to the novel quantization techniques.

---

> ### Author Rebuttal · Authors · 2025-07-30
>
> Thank you for your insightful comments. We have summarized your questions below and provided detailed responses to each.
>
> ### Weaknesses and Questions
> ### Q1. Custom hardware needs stronger motivation and justification for why existing AI accelerators cannot be leveraged.
>
> **Ans:** Codec Avatar represents a transformative paradigm for next-generation social interaction, enabling users in distant locations to communicate in VR as naturally as if they were face-to-face. Leading companies such as Meta and Apple have invested heavily in this technology, recognizing its critical role in the future of spatial computing and virtual telepresence. Meta’s photorealistic avatar research and Apple’s Vision Pro platform both showcase Codec Avatars as foundational to immersive, high-fidelity communication. As these systems evolve into core components of digital life, Codec Avatars are expected to operate as _always-on_ applications, running continuously in the background to support persistent, real-time interaction in social, educational, and professional settings. Their ability to deliver low-latency, lifelike experiences positions them as essential infrastructure for the next era of computing.
>
> Executing Codec Avatar on general-purpose hardware such as GPUs is resource- and power-constrained, resulting in high inference latency of 39.6 ms (lines 107–111), whereas a seamless user experience typically requires 90 FPS [1, 2]. Moreover, GPUs must concurrently support other demanding tasks such as rendering and AI workloads. Therefore, we argue that designing a dedicated hardware accelerator specifically for Codec Avatar applications is both beneficial and necessary to meet real-time performance and user experience requirements.
>
> Furthermore, Codec Avatar models require dedicated hardware due to their unique computational characteristics. The specialized computation flow illustrated in Figure 6(b) necessitates customized computation patterns and scheduling that existing AI accelerators cannot provide. Additionally, conventional AI accelerators are developed without considering VR system integration constraints and compatibility requirements. Most critically, as discussed in Section 2.3, the transposed convolution layers in the Codec Avatar decoder pose a significant challenge due to their inherent structured sparsity (lines 143–147), which substantially reduces the utilization rate and efficiency of the conventional AI accelerator. In contrast, our proposed input-combining accelerator effectively exploits the checkerboard-like structured sparsity in the input activations following the im2col transformation (lines 151–154), leading to a more efficient execution.
>
> [1] Balancing performance and comfort in virtual reality: A study of FPS, latency, and batch values
>
> [2] Survey of motion sickness mitigation efforts in virtual reality
>
> ### Q2. Unclear Benchmarking about the comparison between Snapdragon SoC and NVIDIA GPU.
>
> **Ans:** We apologize for the ambiguity.  The A100 GPU (line 287) was used only for offline PTQ calibration, not runtime benchmarking.
>
> As for model inference, Figure 6 (a) compares three configurations:
>
> | Device                          | Description                                                  |
> |---------------------------------|--------------------------------------------------------------|
> | XR2-2 NPU        | Current commercial baseline (42.04 ms)                       |
> | Baseline Accelerator            | Weight‑stationary dataflow design                            |
> | Our Accelerator                 | Proposed custom hardware accelerator with input‑combining     |
>
> We chose Snapdragon XR2 Gen 2(XR2-2) as the baseline because it is integrated within Meta Quest 3 headset, representing real-world deployment constraints.
>
> The NVIDIA Tesla T4 (lines 289-292) is used to model rendering performance by downclocking to match the capabilities of mobile platforms.
>
> We will clarify these hardware roles in the revised manuscript.
>
> ### Q3.  Reason for the “Co-Optimization” in title.
>
> **Ans:** Our proposed system is explicitly co-designed across both the algorithm and hardware levels, with tightly coupled innovations that are jointly optimized for maximum efficiency.
>
> On the algorithm side, we develop novel smoothing schemes specifically tailored for the Codec Avatar decoder quantization, including both 8-bit and 4-bit. These smoothing strategies are carefully designed to maintain high reconstruction fidelity while significantly reducing computational cost. On the hardware side, we propose a systolic array architecture with a customized multiply-accumulate unit tailored specifically for the quantized decoder model. The design is reconfigurable to support both $8\times 8$-bit and $4\times 4$-bit multiplication. Additionally, each processing element (PE) is optimized to exploit the checkerboard-like activation sparsity that arises from transposed convolution operations after the im2col transformation. Our design is not a generic sparse accelerator, but one that is highly specialized for our quantized Codec Avatar model. It is specifically built to operate efficiently on low-precision (INT8/INT4) data and structured sparse activations.
>
> Moreover, Figure 6(b) illustrates the optimized execution pipeline of Codec Avatar application. Once our quantization algorithm is applied, we can evaluate the inference latency (encoding and decoding) on the proposed custom hardware accelerator. Based on this evaluation, the optimal Codec Avatar working flow can be derived, where the end-to-end latency per frame is bounded by the interval between the two red dashed lines in Figure 6(b). This tight coupling between model quantization and hardware specialization exemplifies our algorithm–hardware co-optimization approach.
>
> Similar way of co-optimizations has been adopted by a lot of other works. For example, in [3, 4, 5], the authors propose FABNet, a hardware-friendly variant of the Transformer architecture, adopting a unified butterfly sparsity pattern for both attention and feed-forward network (FFN) layers. Then, they design a reconfigurable butterfly accelerator, capable of runtime configuration to handle different layers using a unified hardware engine. Our proposed Codec Avatar system follows a similar philosophy. The design of each side was guided and constrained by the other, rather than being independently optimized.
>
> [3] Adaptable Butterfly Accelerator for Attention-based NNs via Hardware and Algorithm Co-design
>
> [4] Packing sparse convolutional neural networks for efficient systolic array implementations: Column combining under joint optimization
>
> [5] SCNN: An accelerator for compressed-sparse convolutional neural networks
>
> ### Q4. What prevents your proposed hardware from further reducing the latency?
>
> **Ans:** The current latency of our proposed hardware accelerator is primarily limited by the size of the systolic array and the capacity of the on-chip buffers. Specifically, we employ a $16\times 16$ systolic array for matrix multiplication, supported by on-chip buffers of 4096 KB for weights, 2048 KB for activations, and 1024 KB for accumulators. While increasing the array size could further reduce latency, it would also result in greater chip area consumption. Under a fixed chip area constraint, this would limit the resources available for other hardware components. Therefore, the array size is carefully selected to achieve a balanced trade-off between performance and energy efficiency.
>
> Additionally, it is important to note that even if the systolic array and memory size are further increased, the per-frame processing latency of ESCA will not be reduced, as the computation is no longer the primary bottleneck. As discussed in Section 4.4 (lines 333-348), the end-to-end frame latency of Codec Avatar application is constrained by Transmission latency. As illustrated in Figure 6(b), the time between the two red dashed lines represents the effective per-frame latency, which equals twice the Transmission latency (10 ms). Further, if the transmission latency gets reduced, the Rendering stage, currently taking 9.5 ms, would become the new bottleneck.
>
> ### Q5. Evaluation regarding the energy consumption comparisons with baselines.
>
> **Ans:** For the custom hardware accelerator, we summarize the estimated energy consumption per frame of model inference in the following table:
>
> | Device   | Precision         | Energy (uJ) |
> |---------|----------------|--------------|
> | Baseline Accelerator | INT8   | 11010.09         |
> | Input-combining Accelerator | INT8   | 3549.64       |
> | Input-combining Accelerator | INT4   |  1653.09      |
>
> As for the XR2-2, there is currently no official disclosure of the absolute power consumption of its on-chip NPU. However, based on available research report [6], typical AI inference power for mobile NPU is approximately 500 mW. Using 0.5 W as a conservative estimate, we perform a back-of-the-envelope calculation by multiplying the power with the measured inference latency to estimate energy consumption:
>
> | Device   | Precision         | Energy (uJ) |
> |---------|----------------|--------------|
> | XR2-2 NPU | FT32   | 19800         |
> | XR2-2 NPU | INT8   | 9250       |
> | XR2-2 NPU | W4A8   |  8500      |
>
> Notably, the W4A8 reflects the current capability of the Qualcomm AI Hub, which supports 4-bit weights and 8-bit activations [7]. Our ESCA accelerator achieves significantly lower energy consumption compared to both the baseline accelerator and the commercial XR2-2 SoC.
>
> [6] Deep Learning on Mobile Devices With Neural Processing Units
>
> [7] Qualcomm AI Hub. (n.d.). Documentation. Qualcomm. app.aihub.qualcomm.com/docs/

---

> ### Author Response · Authors · 2025-08-08
>
> Dear Reviewer,
>
> We hope you’re doing well. We kindly note that today is the final day for the author–reviewer discussion phase for our submission. If you have any remaining feedback or questions regarding our rebuttal, we would be grateful to hear them.
>
> Thank you for your time and consideration throughout this process. We truly appreciate your insights and engagement.
>
> Best,
>
> Authors

---

> > ### Comment · Reviewer_p8Jg · 2025-08-08
> >
> > The rebuttal argues that a dedicated hardware accelerator is necessary because Codec Avatar is a "transformative paradigm" and a "core component of digital life." While these are strong motivations, they don't directly answer the question about why existing AI accelerators that already support sparse and mixed-precision operations are inadequate. The response mentions that conventional accelerators can't handle the "unique computational characteristics" or "specialized computation flow" of Codec Avatar models, but it doesn't provide a direct comparison. The rebuttal also states that the accelerator is needed to exploit the "checkerboard-like structured sparsity" of transposed convolutions, which is a key claim, but again, it lacks a comparison to other sparse-aware accelerators. What is fundamentally different for a checkerboard pattern that shows up in many super-resolution workflows that a TPU's systolic array [1] and its dedicated compiler stack cannot fundamentally support?
> >
> > I maintain my concern regarding the insufficient justification for the custom hardware, particularly the lack of a direct comparison with the performance achievable in software on existing accelerators. However, I will not stand against the paper's acceptance if the other opinions are in favor of it.
> >
> > [1] https://cloud.google.com/tpu/docs/system-architecture-tpu-vm

---

> ### Author Response · Authors · 2025-08-09
>
> We appreciate the reviewer’s thoughtful comments.
>
> ### Performance Comparison and Baseline Evaluation.
>
> The reviewer correctly notes that conventional accelerators can execute Codec Avatar inference. However, our key contribution lies in demonstrating that existing solutions fail to meet real-time VR requirements due to efficiency limitations. Section 4.4 (lines 327-333) provides a direct quantitative comparison: our input-combining hardware accelerator achieves 3.36$\times$ speedup over a baseline weight-stationary systolic array, reducing INT8 decoder latency from 42.04 ms to 12.51 ms. This reduction is a performance gain essential for immersive VR applications.
>
> ### Distinction from Existing Sparse Accelerator.
>
> Most existing AI accelerators with support for sparse operations [1, 2] primarily target **unstructured weight sparsity** from pruning techniques applied to reduce model complexity for edge deployment [3]. Their optimization focuses on eliminating multiply-by-zero operations in pruned weight matrices. Our approach addresses a fundamentally different sparsity type: **structured activation sparsity** inherent to transposed convolution operations in Codec Avatar decoders. This sparsity manifests as fixed checkerboard patterns in input activations after the im2col transformation (lines 151-154), not due to artificial model pruning. Critically, our model weights remain fully dense, making traditional pruning-oriented accelerators ineffective. The pruning-oriented accelerators would perform similarly to our baseline.
>
> ### Comparison with Super-Resolution Accelerators
>
> We acknowledge related work in SR accelerators, including designs for pruned deconvolution weight sparsity [4], input spatial sparsity [5], and conditional computation [6]. However, these architectures either skip zero weights post-pruning or reduce computation through selective spatial region processing. Neither approach exploits the fixed checkerboard activation patterns emerging from the transposed convolution after im2col transformation. Our input-combining accelerator specifically targets systolic array GEMM execution with dense weights while compressing and scheduling sparse activation.
>
>
> [1] Packing Sparse Convolutional Neural Networks for Efficient Systolic Array Implementations: Column Combining Under Joint Optimization
>
> [2] SASCHA—Sparsity-Aware Stochastic Computing Hardware Architecture for Neural Network Acceleration
>
> [3] Learning both weights and connections for efficient neural networks
>
> [4] SDCNN: An Efficient Sparse Deconvolutional Neural Network Accelerator on FPGA
>
> [5] SRNPU: An Energy-Efficient CNN-Based Super-Resolution Processor With Tile-Based Selective Super-Resolution in Mobile Devices
>
> [6] ESSR: An 8K@30FPS Super-Resolution Accelerator With Edge Selective Network

---

### Official Review · Reviewer_6fss · 2025-07-05

**Clarity:** 3
**Significance:** 3
**Originality:** 3
**Rating:** 4
**Confidence:** 3

**Summary:**

The paper introduces a framework designed to optimize photorealistic codec avatars for real-time execution on resource-constrained AR/VR devices. It focuses on post-training quantization techniques and a custom hardware accelerator to reduce the computational burden and latency associated with decoding. Specifically, the proposd ESCA integrates several techniques, including input channel-wise activation smoothing (ICAS), facial-feature-aware smoothing (FFAS), UV-weighted hessian-based weight quantization (UV-Weighted PTQ), and a hardware accelerator tailored to the specific needs of AR/VR devices. Experimental results demonstrate the effectiveness of the proposed method.

**Questions:**

1. What is reason for the flickering artifacts around the edges between the hair and face in the final rendering?
2. How do the proposed quantization techniques and the custom hardware accelerator contribute to the overall latency reduction individually?
3. How sensitive is the proposed method to variations in the quality of the UV map?

**Ethical Concerns:**

["NO or VERY MINOR ethics concerns only"]

**Final Justification:**

The rebuttal addressed my concerns well, and I would like to keep the postive rating.

**Limitations:**

Yes.

**Paper Formatting Concerns:**

No.

**Quality:**

3

**Strengths And Weaknesses:**

Strengths:
1.  It is interesting to integrate the algorithmic optimizations with a custom hardware accelerator for reducing latency in codec avatar rendering.
2. The proposed techniques, e.g., ICAS, FFAS, UV-PTQ, are meaningful and well-motivated. They together provide an efficient way to reduce computational demands while maintaining high fidelity in avatar reconstruction, especially for critical facial features.
3. The proposed method demonstrates obvious improvements in both visual quality and latency.
4. The end-to-end pipeline runs at over 100 FPS on resource-constrained platforms. This is valuable for real-time AR/VR applications.

Weaknesses:
1. I carefully check the video results, and find that there are noticeable flickering artifacts (around the edges between the hair and face) in the final rendering, which could negatively impact the user experience in immersive environments.
2. The contribution of each part of the pipeline in reducing latency is not very clear to me. More explaination is expected.
3. Only the MultiFace dataset and a small number of avatars are used for evaluation. More diverse datasets and real-world scenarios are expected. This would help demonstrate the robustness and adaptability of the proposed method.
4. I am also curious about how sensitive the proposed method is to the quality of the UV map.

Overall, while I am not an expert in this specific task, I find the proposed method both interesting and meaningful. The combination of algorithmic and hardware optimizations is promising. Further clarification and broader evaluation would strengthen the paper. I would like to raise my score based on the authors' feedback.

---

> ### Author Rebuttal · Authors · 2025-07-30
>
> Thank you for your insightful comments. We have summarized your questions below and provided detailed responses to each.
>
> ### Weaknesses and Questions
> ### Q1. What is the reason for the flickering artifacts around the edges between the hair and face in the final rendering?
>
> **Ans:** Thanks for pointing this out, as discussed in lines 38-42, flickering in low-bit quantized models stems from unstable quantization of activation outliers. Our decoder's transposed-convolution layers process stochastic latents without normalization, creating heavy-tailed activation distributions (Figure 1). Hair-face boundaries are particularly challenging due to alternating dark/bright UV texels, generating the largest activation spikes that dominate the quantization range and manifest as flicker.
>
> ESCA explicitly addresses this through Input Channel-wise Activation Smoothing (ICAS) which suppresses outlier-driven quantization error, and Facial-Feature-Aware Smoothing & UV-Weighted Quantization which strategically allocates precision to perceptually critical regions (eyes, mouth)(Eq. 4). This design leverages psychovisual research showing viewers are sensitive to facial artifacts [1,2]. While minor artifacts may remain at the edge under aggressive 4-bit quantization, we substantially improve stability where it matters most. Additionally, our method achieves +0.39 FovVideoVDP improvement over the best 4-bit baseline (Table 1). This metric specifically captures spatio-temporal artifacts, providing strong quantitative evidence of reduced flickering.
>
> Supplementary video and Figure 7 show baseline has severe flickering in key facial regions. ESCA limits flickering mainly to hair-face boundaries while maintaining stable, high-fidelity details elsewhere. Future work could refine training algorithms to further reduce these artifacts.
>
> [1] You look blocky, is everything alright? Influence of video distortions on facial expression recognition & quality
>
> [2] InternVQA: Advancing Compressed Video Quality Assessment with Distilling Large Foundation Model
>
>  ### Q2. Clarify latency reduction of each pipeline component.
>
> **Ans:** Figure 3(b) illustrates the end to end execution flow, where all five components, Sensing, Encoding, Transmission, Decoding, and Rendering, are executed sequentially to process each sample. As detailed in Section 4.4 (lines 335-339), the latencies for Sensoring, Transmission, and Rendering are 1 ms, 5 ms, and 9.5 ms, respectively. Meanwhile, as discussed in Section 2.1 (lines 107-110) and shown in Figure 6(a), the inference times for Encoding and Decoding are 13.8 ms and 25.8 ms on Snapdragon XR2 Gen 2 (XR2-2) SoC. Consequently, the total latency per frame would be 55.1 ms for the original pipeline, corresponding to a throughput of 18.15 frames per second (FPS).
>
> In contrast, Figure 6(b) illustrates the optimized pipeline, which introduces two major changes. First, the pipelines of two users are processed in an overlapped fashion to maximize throughput. The introduction of the accelerator provides an additional hardware resource dedicated to executing Encoding and Decoding, while the rendering operations can be assigned to the GPU to achieve maximum parallelism. In addition, Transmission and Decoding of each user are executed concurrently. At the same time, the decoder waits for corresponding transmission completion since it requires the other user's encoder output as input. This reduces end-to-end latency to 10 ms (2 $\times$ 5 ms transmission time), achieving 100 FPS with significant latency and throughput improvements.
>
> ### Q3. Evaluation use only MultiFace dataset.
> **Ans:** To the best of our knowledge, MultiFace is the only public Codec-Avatar dataset with pretrained models, enabling our post-training quantization experiments. Alternative datasets (Goliath [3], Tele-ALOHA [4]) provide only raw data requiring extensive training unfeasible within rebuttal timeframes [13], We will include additional evaluations in the final version of the paper once a suitable dataset with pretrained models becomes available.
>
> Although our study uses a single dataset, MultiFace offers a rigorous evaluation setting: 65TB with 13,000+ 2K frames per subject across 65 expressions and multiple viewpoints. The comprehensive coverage of extreme poses, occlusions, and high-frequency details creates a demanding testbed validating low-precision decoder performance. Current MultiFace evaluation represents the most scientifically sound approach given practical constraints while comprehensively validating quantization robustness.
>
> While we did not evaluate ESCA on additional datasets, we demonstrate its applicability to image super-resolution tasks, which aim to enhance the resolution of input images. For this task, although our UV-weighted PTQ cannot be extended to this domain due to the absence of UV coordinates, our smoothing can be used to facilitate the quantization over these super-resolution tasks.
>
> We compare our method against GPTQ, the best-performing baseline from Codec-Avatar quantization, across five commonly used super-resolution benchmarks [5, 6, 7, 8, 9]. Our quantization experiments used the SwinIR [10] super-resolution model, which was pretrained on the combined DIV2K [11] and Flickr2K [12] datasets.
>
> | Method   | Bit | Set5 PSNR | Set5 SSIM | Set14 PSNR | Set14 SSIM | B100 PSNR | B100 SSIM | Urban100 PSNR | Urban100 SSIM | Manga109 PSNR | Manga109 SSIM |
> |----------|-----|-----------|-----------|------------|------------|-----------|-----------|----------------|----------------|----------------|----------------|
> | SwinIR R | 32  | 31.13   | 0.8967    | 27.45    | 0.7867     | 26.37   | 0.7434    | 25.15        | 0.7993         | 29.60        | 0.9115         |
> | ESCA     | 4   | 28.63   | 0.8310    | 25.85    | 0.7277     | 25.29  | 0.6885    | 22.87        | 0.6852         | 26.79        | 0.8088         |
> | GPTQ     | 4   | 27.90   | 0.7729    | 25.41    | 0.6776     | 24.74  | 0.6338    | 22.68       | 0.6462         | 25.94        | 0.7555         |
>
> ESCA achieves superior 4-bit quantization performance compared to GPTQ across all five super-resolution benchmarks, with PSNR improvements ranging from 0.19 dB (Urban100) to 0.85 dB (Manga109) and SSIM improvements ranging from 0.039 (Urban100) to 0.0581 (Set5).
>
> [3] Codec avatar studio: Paired human captures for complete, driveable, and generalizable avatars
>
> [4] Tele-Aloha: a low-budget and high-authenticity telepresence system using sparse RGB cameras
>
> [5] Low-complexity single-image super-resolution based on nonnegative neighbor embedding
>
> [6] On single image scale-up using sparse-representations
>
> [7] A database of human segmented natural images and its application to evaluating segmentation algorithms and measuring ecological statistics
>
> [8] Single image super-resolution from transformed self-exemplars
>
> [9] Sketch-based manga retrieval using manga109 dataset
>
> [10] Swinir: Image restoration using swin transformer
>
> [11] Ntire 2017 challenge on single image super-resolution: Methods and results
>
> [12] Enhanced deep residual networks for single image super-resolution
>
> [13] High-quality joint image and video tokenization with causal VAE
>
> ### Q4.  How sensitive is the proposed method to variations in the quality of the UV map?
>
>
> **Ans:** To quantitatively assess UV map quality sensitivity, we conducted an ablation study by randomly dropping UV importance weights from 10% to 50% before 4-bit PTQ processing. This simulates progressively degraded UV priors. The table below shows the FovVideoVDP results on the three views renderings as we degrade the UV map quality.
> | UV Map Dropout Ratio               | Front View | Left View  | Right View  |
> |-----------------------|------------|--------------|--------------|
> | 50%           | 5.7828     | 4.9034       | 4.9084       |
> | 40%           | 5.8366     | 4.9718       | 4.9540       |
> | 30%           | 5.8421     | 4.9739       | 4.9549       |
> | 20%           | 5.8401     | 4.9756       | 4.9560       |
> | 10%             | 5.8454     | 4.9754       | 4.9582       |
> | ESCA  | 5.8541     | 4.9795       | 4.9605       |
> The ablation study shows that ESCA maintains robustness despite declining UV map quality.
>
> ### Q5.  Latency contributions of the quantization and hardware accelerator?
>
> **Ans:** The impact of the custom hardware accelerator on overall latency reduction is discussed in Section 4.4 (lines 327–337) and illustrated in Figure 6(a). A summary of the results is provided below:
>
> | Device   | Precision         | Latency (ms) |
> |---------|----------------|--------------|
> | Baseline Accelerator | INT8   | 45.09         |
> | Input-combing Accelerator | INT8   | 15.56         |
> | Input-combing Accelerator | INT4   |  6.18      |
>
> For the contribution of quantization techniques, we only report in Section 2.1 (lines 107-111) that the inference latency of the full-precision Codec Avatar model on XR2-2 is 39.6 ms. Due to space constraints, the performance of quantized models was not included in the paper. We provide the detailed results here:
>
>
> | Device   | Precision         | Latency (ms) |
> |---------|----------------|--------------|
> | XR2-2 | FT32   | 39.6         |
> | XR2-2 | INT8   | 18.5         |
> | XR2-2 | W4A8   |  17.0      |
>
> W4A8 reflects current Qualcomm AI Hub capability limits (4-bit weights, 8-bit activations only) [14]. Overall, the proposed quantization techniques and custom hardware accelerator each contribute to a multi-fold reduction in inference latency.
>
> [14] Qualcomm AI Hub. (n.d.). Documentation. Qualcomm. app.aihub.qualcomm.com/docs/

---

> ### Author Response · Authors · 2025-08-08
>
> Dear Reviewer,
>
> We hope you’re doing well. We kindly note that today is the final day for the author–reviewer discussion phase for our submission. If you have any remaining feedback or questions regarding our rebuttal, we would be grateful to hear them.
>
> Thank you for your time and consideration throughout this process. We truly appreciate your insights and engagement.
>
> Best,
>
> Authors

---

> > ### Comment · Reviewer_6fss · 2025-08-09
> >
> > Thank the authors for their detailed reply. The rebuttal addressed my concerns, and I will keep the postive rating.

---

### Author Response · Authors · 2025-08-05

We appreciate your thoughtful review and the insights you provided. In our rebuttal, we aimed to address all the issues you raised. If there are aspects of our response that remain unclear, or if you'd like to discuss any point further, please feel free to follow up. Your feedback is important to us, and we welcome any additional discussion.

---

### Author Response · Authors · 2025-08-07

We appreciate your thoughtful reviews and have addressed all concerns in our rebuttal posted 7 days ago.
With the discussion period closing August 8th, we’d welcome any feedback on our responses to ensure we’ve adequately addressed your questions.
If our responses need clarification or additional information, please let us know. We value your continued input.

---

### Decision · Program_Chairs · 2025-09-17

**Decision:**

Accept (poster)

**Comment:**

This paper was reviewed by four experts in the field, with 1 Boarderline Reject, 2 Boarderline Accept and 1 strong Accept. After reviewing the rebuttal and discussion, most reviewers are inclined to acceptance. Reviewers appreciate that this paper proposes an interesting solution for light-weight Codec Avatar, with reasonable technical contributions like post-training quantization and a custom hardware accelerator. Thus, AC sided with the positive reviewers. AC highly urges the authors to go through the detailed comments carefully to polish the writing and tune down the vague illustration, and provide extra experimental details and  qualitative video results, so as to ensure the final acceptance.